# Current Landscape of Immune Checkpoint Inhibitors for Metastatic Urothelial Carcinoma: Is There a Role for Additional T-Cell Blockade? [note 1]

**DOI:** 10.3390/cancers16010131

**Published:** 2023-12-27

**Authors:** Vanessa Ogbuji, Irasema C. Paster, Alejandro Recio-Boiles, Jennifer S. Carew, Steffan T. Nawrocki, Juan Chipollini

**Affiliations:** 1College of Medicine, University of Arizona, Tucson, AZ 85724, USA; cogbuji@arizona.edu (V.O.); sa203699@atsu.edu (I.C.P.); snawrocki@arizona.edu (S.T.N.); 2Department of Medicine, The University of Arizona Cancer Center, Tucson, AZ 85724, USA; areciomd@arizona.edu (A.R.-B.); jcarew@arizona.edu (J.S.C.)

**Keywords:** urothelial carcinoma, immune checkpoint inhibitor, bladder cancer, PD-1, CTLA-4

## Abstract

**Simple Summary:**

Immune checkpoint inhibition has played a significant role in the treatment landscape of urothelial carcinoma due to the highly immunogenic tumor microenvironment. However, there remains a suboptimal response to therapy by patients due to resistance to therapy. New paradigms of treatment using PD-1, PD-L1, and CTLA-4 inhibitors are vital for addressing resistance to treatment. In this review, we describe preclinical and clinical literature on immune checkpoint inhibitors in urothelial carcinoma. Specifically, we propose the use of combinatorial therapy of already FDA-approved immune checkpoint inhibitors to boost the response of the immune system. There are currently no FDA-approved therapies that combine PD-1/PD-L1 inhibitors with CTLA-4 inhibitors despite approvals in other solid tumors. Successful implementation of combined PD-1/PD-L1 and CTLA-4 inhibitors could significantly improve patient response. Here, we discuss preclinical and clinical research findings to broaden the knowledge of immune checkpoint inhibitors for urothelial carcinoma therapy.

**Abstract:**

Urothelial carcinoma (UC) is the most common form of bladder cancer (BC) and is the variant with the most immunogenic response. This makes urothelial carcinoma an ideal candidate for immunotherapy with immune checkpoint inhibitors. Key immune checkpoint proteins PD-1 and CTLA-4 are frequently expressed on T-cells in urothelial carcinoma. The blockade of this immune checkpoint can lead to the reactivation of lymphocytes and augment the anti-tumor immune response. The only immune checkpoint inhibitors that are FDA-approved for metastatic urothelial carcinoma target the programmed death-1 receptor and its ligand (PD-1/PD-L1) axis. However, the overall response rate and progression-free survival rates of these agents are limited in this patient population. Therefore, there is a need to find further immune-bolstering treatment combinations that may positively impact survival for patients with advanced UC. In this review, the current immune checkpoint inhibition treatment landscape is explored with an emphasis on combination therapy in the form of PD-1/PD-L1 with CTLA-4 blockade. The investigation of the current literature on immune checkpoint inhibition found that preclinical data show a decrease in tumor volumes and size when PD-1/PD-L1 is blocked, and similar results were observed with CTLA-4 blockade. However, there are limited investigations evaluating the combination of CTLA-4 and PD-1/PD-L1 blockade. We anticipate this review to provide a foundation for a deeper experimental investigation into combination immune checkpoint inhibition therapy in metastatic urothelial carcinoma.

## 1. Introduction: Urothelial Carcinoma

According to a systematic review conducted by GLOBOCAN, there were 573,000 new BC cases and 213,000 deaths worldwide in 2020 [1]. BC is four times more common in men compared with women [2,3]. Urothelial carcinoma (UC), also known as transitional cell carcinoma, is the most common histology type with risk factors including smoking, toxins, age, gender, race, chronic bladder infections, and history of bladder cancer [2]. Squamous cell carcinoma and adenocarcinoma histology have a lower incidence [2]. UC affects a significant percentage of the older population, and the incidence rate continues to rise every year. UC can be divided into three categories: non-muscle-invasive bladder cancer (NMIBC), muscle-invasive bladder cancer (MIBC), and metastatic UC, with metastatic UC displaying high mortality rates [4] (Figure 1).

The definitive treatment for MIBC is radical cystectomy with or without neoadjuvant chemotherapy [6]. Numerous agents are approved for adjuvant treatment of UC including platinum-based chemotherapy, radiation, and more recently immune checkpoint inhibitors (ICIs) [6]. Platinum-based chemotherapy has been the standard of care for metastatic UC [6]. However, many patients relapse after chemotherapy or are unable to tolerate platinum chemotherapy due to medical comorbidities [6].

Newer agents, most notably ICIs, have been approved for the treatment of both NMIBC and MIBC. ICIs target ligands that lead to inactivation and apoptosis of T-cells, specifically cytotoxic T-cells [7]. Patients who are treated with ICIs for bladder cancer are administered programmed cell death protein 1 (PD-1) or programmed death-ligand 1 (PD-L1) antibody therapy at various stages of the disease. However, ICIs in metastatic UC do not achieve a complete response rate in most patients [4]. Multiple immune checkpoint ligands play a part in immune evasion during bladder tumorigenesis [8]. Understanding the molecular pathways of the ligands these inhibitors act on could be important in increasing the response rate of ICI therapies.

In the bladder tumor microenvironment, some factors promote tumor evasion of the immune system using immune checkpoint-related molecules. The mechanism of ICIs is to abrogate these interactions and prevent apoptosis of T-cells. Due to the multifactorial aspects of immune checkpoint molecules, a combinatorial approach using ICIs could target different pathways of immune evasion and result in a more robust response. Given the challenge of tumor resistance to ICI treatment after conventional chemotherapy, there remains a need for novel agents that may bolster the immunomodulating response of currently available therapies. In this narrative review of the literature, we discuss the current landscape of FDA-approved checkpoint inhibitors in the management of advanced UC and expound on the scientific rationale for combination therapy in the form of PD-1/PD-L1 blockade and CTLA-4.

### 1.1. Tumor Microenvironment in Urothelial Carcinoma

In UC, the tumor microenvironment (TME) consists of stroma, connective tissue, blood vessels, and immune cells. These different cell types provide nutrition and support to the tumor parenchyma. Communication between the tumor cells and their environment can alter the TME and promote metastasis [9]. Among the cells in the microenvironment are the stromal cells which make up the connective tissue and extracellular matrix (ECM) of the environment. In UC, cancer-associated fibroblasts (CAFs) are the most common stromal cell types present in the TME [10]. Activated CAFs express vimentin, α-smooth muscle actin, platelet-derived growth factor receptor, and other markers, all of which aid in the remodeling of the ECM [10]. There has been controversy over the exact role of CAFs in cancer biology, and due to the heterogeneity of these cells, they could either promote or inhibit tumor growth. Bladder cancer research has found that the frequency of fibroblasts is increased in UC compared with normal uroepithelium [10]. Moreover, a clinical study with 344 bladder cancer patients found that dominant expression of CAF markers is negatively correlated with 5-year survival and positively correlated with muscle invasiveness of cancer [11,12]. While the mechanism behind this is still being explored, one possible explanation would be the role CAFs have on immune cells in the TME. Stromal cells in the TME can release soluble factors that alter cell surface proteins and suppress the immune system; for example, they can regulate the expression of PD-L1 on tumor cells [9]. Factors regulated by CAFs can lead to the release of anti-inflammatory cytokines such as TGFβ. TGFβ can then induce the production of immunosuppressive T-regulatory cells [13]. The desmoplasia that is promoted by CAFs and TGFβ can also contribute to the exclusion of immune cells in the tumor parenchyma and localize them to the stroma [14,15]. Wang. et al. found that bladder cancer samples with high T-cell infiltration and low stroma-related gene expression had longer survival [14]. These modulations by CAFs all impact the immune microenvironment of UC, especially the tumor-infiltrating lymphocytes (TIL) (Figure 2).

Additionally, CAFs increase the expression of BCL2 through estrogen receptor alpha (Erα) signaling in vivo [16]. BCL2 is a key antiapoptotic gene that prevents the activation of proapoptotic proteins like BAK/BAX [16]. In this way, CAFs promote the growth of UC by constitutively activating antiapoptotic proteins. The constitutive expression of BCL2 in bladder cancer also has downstream effects on the immune cells in the TME.

### 1.2. Tumor Mutational Burden in Bladder Cancer

Infiltration of T-cells into the TME is indicative of cellular immune response to tumor neoantigens. UC has been shown to have a robust immune response, which is largely based on its high tumor mutational burden (TMB). Based on the Cancer Genome Atlas, bladder cancer has high somatic mutation rates along with lung and skin cancer [17]. High somatic mutation and TMB allow increased circulation of neoantigens that can be used to prime cellular immunity in the host [17]. Identification of frequent mutations has allowed several institutions to create molecular subgrouping of UC. Particularly, MIBC has been categorized into five subgroups: luminal papillary (LumP), luminal non-specified (LumNS), luminal unstable (LumU), stroma-rich, basal/squamous (Ba/Sq), and neuroendocrine-like (NE-like) subtypes [10]. The LumU and basal/squamous subtypes have the highest level of genomic instability due to somatic mutations [18]. The most common genes mutated in this subtype are the ones that encode the apolipoprotein B mRNA editing catalytic polypeptide-like family of proteins (APOBEC) [10,19]. The APOBEC-specific mutations seen in bladder cancer are commonly known to be in the telomerase reverse transcriptase (TERT) promoter region. Furthermore, 70–80% of bladder cancers contain this mutation [20]. TERT is the catalytic subunit of the telomerase holoenzyme and it is responsible for preventing telomere shortening [10]. During embryonic development, TERT activity is high, but in somatic tissues, the expression levels decrease. In somatic tissues, TERT overexpression promotes cell division and compromises cell cycle regulation. The pathological expression of TERT allows for the immortality of tumor cells. Among the patients with UC, those with high expression of TERT have a lower disease survival rate [21]. Other mutations commonly seen in UC include TP53, RB1, and EGFR [19]. UC bladder cancer has a high TMB, suggesting an increase in neoantigen presentation to the cellular immune system. Therefore, the immune environment within UC consists mostly of T-lymphocytes.

### 1.3. Tumor-Infiltrating Immune Cells in Bladder Cancer

Within the immune microenvironment of UC, there are several immune cells present, and the major effector immune cells are the tumor-infiltrating lymphocytes (TILs). Among TILs, cytotoxic T-cells (CD8^+^ T-cells) are the most important to mounting a response against the neoantigens presented on the tumor [22]. Studies have shown a positive correlation between the amount of CD8^+^ T-cells present in tumor tissue and the patient’s response to immunotherapy [22]. Due to the effector function of CD8^+^ T-cells, it is conceivable that most immune evasion tactics from the tumor are directed toward these cell types. CD8^+^ T-cells carry out their functions via the use of catalytic enzymes, which promote lysis of the cells they target [10]. Additionally, CD8^+^ T-cells release other cytokines that prime immune cells toward the tumor microenvironment. These cells are incredibly important in the restraint of tumor growth and are key therapeutic targets for immunotherapy. There are cell receptors that are present on CD8^+^ T-cells that can be upregulated or downregulated by the tumor cells. This modification of receptors helps the tumor evade the TILs. For example, immune checkpoints are cell-surface proteins that are used by tumor cells in the TME. CD8^+^ T-cells are presented with peptide antigens on MHC-I molecules present on antigen-presenting cells (APC). The binding of the T-cell receptor to the antigen is not enough to activate the CD8^+^ T-cells; other signals like the binding of CD80 on APC to CD28 on T-cells and the presence of cytokines, like IFNγ, are needed to fully activate the CD8^+^ T-cells [23]. While CD8^+^ T-cells are important in the direct killing of tumor cells, helper T-cells (CD4^+^ T-cells) support the role of CD8^+^ T-cells. The Th1 CD4^+^ T-cells pathway leads to the release of inflammatory cytokines, particularly IFNγ. IFNγ has multiple effects on different cells including activation of CD8^+^ T-cells and macrophages [24]. In this manner, CD4^+^ T-cells are necessary for the differentiation of immune cells in the TME. Studies in bladder cancer cell lines have found that depletion of CD4^+^ T-cells in the TME increases tumor size and dampens the effect of immune checkpoint inhibition [25]. Immunotherapy targets multifactorial aspects of the signaling pathway that leads to CD8^+^ T-cell and CD4^+^ T-cell activation. Moreover, there are other immune cells in the TME that impact the immune microenvironment in UC.

Other cells in the TME that contribute to the immune microenvironment are macrophages. They decrease the efficacy of cancer therapy including chemotherapy and immunotherapy [26]. Macrophages are phagocytic cells that differentiate from circulation monocytes to become tissue-resident cells. In normal tissues, macrophages act as part of the innate system to fight foreign pathogens through phagocytosis. In the adaptive immune system, macrophages can present antigens to activated T-cells to modulate cellular immunity. Additionally, macrophages play a role in wound healing and coordinate anti-inflammatory cytokines in the body after infection. A key feature of macrophages is their ability to differentiate into one of two phenotypes in the presence of the appropriate cytokines and gene expression [26]. The M1 phenotype is pro-inflammatory and is activated in the presence of cytokines like IFNγ, TNFα, IL-6, and IL-1 [26]. The endpoint of M1 function is to fight infection and tumors and prevent wound healing. Conversely, the M2 phenotype is anti-inflammatory and mediates the wound-healing process in the body [25]. The M2 phenotype is activated in the presence of cytokines like IL-4, IL-3, and IL-10 [25]. M2 macrophages secrete mediators like IL-10 and TGFβ to dampen the inflammatory process [26].

During tumorigenesis, the macrophages present in the TME are referred to as tumor-associated macrophages (TAM). These can either be resident in the tissue at the time of neoplasia or they are recruited from circulating monocytes into areas of hypoxia [10]. Tumor cells secrete cytokines that favor the differentiation of TAM into the M2 phenotype, thereby dampening inflammation and potentiating tumor growth. In bladder cancer, TAMs found in tumor samples are predominantly of the M2 phenotype and are commonly present in high-grade diseases [10]. Several cytokines are secreted by these M2 macrophages, including IL-10 and TGFβ. IL-10 inhibits the function of APCs, they suppress the maturation of intratumoral dendritic cells and reduce IL-12 production [27]. All these pathways that are disrupted are needed in the activation of CD8^+^ T-cells in the TME; therefore, M2 macrophages can dampen the cytotoxic T-cell response needed in UC. The TME of UC is a complex system with multifactorial cells playing different parts in immune evasion, evasion of apoptosis, and cell immortality to aid in the growth and sustenance of the tumor. Understanding these factors is important for the development of novel therapeutic strategies, including the use of ICIs to treat UC.

## 2. Immune Checkpoints in Urothelial Carcinoma: Mechanism of Action

PD-L1 is a ligand that can be expressed on tumor cells as well as tumor-associated immune cells. PD-L1 binds to PD-1, a receptor expressed on effector T-cells. In normal uroepithelium, PD-L1 only binds to PD-1 when the immune system has been stimulated for a prolonged period. Moreover, new research has shown that PD-L1 is also expressed on APCs and binds to the ligand CD80 [28]. Cytotoxic T-lymphocyte antigen-4 (CTLA-4) is a protein that also interacts with CD80. While CTLA-4, PD-L1, and PD-1 all work as “brakes” to the immune system, they do so at different stages of T-cell activation. CTLA-4 has been found to act at the priming stage when T-cells first differentiate into effector states [29]. PD-1 and PD-L1 are more important during the elimination phase where they encounter the antigen in the periphery [29]. Tumor cells have adapted these ligands and use them to escape the immune system during tumorigenesis.

### 2.1. Program Death 1 (PD-1)

PD-1 is a major cell surface checkpoint receptor present on T-cells in the TME of UC. In tumorigenesis, PD-1 expression is greatly increased on activated T-cells, B-cells, natural killer cells, and myeloid-derived cells to decrease the proliferation and activation of these immune cells [30]. Studies have shown that PD-1 upregulation in UC is more prominent in CD8^+^ T-cells versus CD4^+^ T-cells, although its expression is still overexpressed in both cell types [31]. PD-1 binds to two ligands, PD-L1 and PD-L2. However, PD-L2 is rarely present in high concentrations. PD-1 primarily asserts its actions by binding to PD-L1, which leads to apoptosis of T-cells [30]. The downstream effect of PD-1 signaling is the downregulation of the PI3k/AKT pathway that shuts down cytokine secretion of T-cells as well as the cytolytic effects of CD8^+^ T-cells [32]. Importantly, tumor cells expressing high PD-L1 levels can escape apoptosis due to immune avoidance [33]. Currently, there are two PD-1 immune checkpoint inhibitors approved by the FDA for metastatic UC: pembrolizumab and nivolumab.

### 2.2. Programmed Death-Ligand 1 (PD-L1)

PD-L1 is a receptor expressed both on the surface of tumor cells and host immune cells in the TME. The binding of PD-L1 to PD-1 downregulates T-cells. Additional research has found that PD-L1 also binds to CD80 on dendritic cells [27]. In vitro experiments conducted by Mayoux et al. showed that PD-L1 colocalizes with CD80, thereby sequestering it from binding to CD28 [28]. In this way, PD-L1 can induce anergy of T-cells by inhibiting the appropriate signaling needed for activation. PD-L1 upregulation can be induced in the TME of UC in response to cytokines in the TME, oncogenic alterations, and hypoxia [27].

During inflammation, IFNγ acts as a pro-inflammatory cytokine as it drives T-cell proliferation and activation of other immune cells like natural killer cells and M1 macrophages [34]. In the TME of UC, IFNγ has been shown to play a role in the upregulation of PD-L1 during in vitro cell line studies. One particular study showed that when dendritic cells are co-cultured with IFNγ, it leads to increased expression of PD-L1 [7]. IFNγ binds to interferon-gamma receptor (IFNGR), leading to activation of the JAK/STAT pathway via STAT1 [34]. The result of this signaling cascade is the activation of the transcription factor interferon-responsive factor 1 (IRF1). IRF1-deficient mice displayed decreased tumor growth in colon cancer mouse models [35]. Therefore, IRF1 is a key mediator of IFNγ-induced expression of PD-L1 in the TME (Figure 3).

UC exhibits a high rate of somatic mutations and these oncogenic alterations may significantly contribute to PD-L1 upregulation. The p38/MAPK signaling cascade plays a positive role in PD-L1 expression on dendritic cells in bladder cancer [36]. Epigenetic regulations through DNA methylation and histone modification are another potential mediator for increased PD-L1 expression. A recent study conducted by Zhang et al. explored the role of an epigenetic protein called WDR5 in immune evasion via PD-L1 expression in bladder cancer [37]. WDR5 is a histone presenter that forms a complex with the MLL1-MLL4 methyltransferase. This complex plays a vital role in chromatin remodeling, transcriptional activation of genes, and histone methylation in bladder cancer [38]. Using genomic data from the cancer genome atlas (TCGA), they found that WDR5 is positively correlated with the expression of PD-L1 in different bladder cancer subtypes. They also found that competitive inhibition of WDR5 led to a decrease in PD-L1 expression in the TME of bladder cancer cell lines, even in the presence of IFNγ [37]. Inhibiting WDR5 led to a decrease in PD-L1 expression at the mRNA level by decreasing RNA polymerase II levels in the PD-L1 promoter region [37]. These experiments demonstrated that oncogenic alterations in bladder cancer also impact the upregulation of PD-L1 in the TME. In patients with UC, higher PD-L1 expression has been associated with higher staging and lower chances of recovery [28].

### 2.3. Cytotoxic T-Lymphocyte-Associated Protein-4 (CTLA-4)

The first immune checkpoint to be clinically targeted for cancer treatment was CTLA-4. CTLA-4 is expressed on activated CD4^+^ T-cells and CD8^+^ T-cells with the former having higher levels of expression [30]. It is also expressed on certain subsets of T-regulatory cells in the TME [28]. CTLA-4 has a similar function to PD-L1 expressed on APCs and binds CD80 with approximately ten times more affinity compared with CD28 [29,30]. Without CD80 and CD28 binding, the co-stimulation of T-cell activation is not achieved [30]. In this way, CTLA-4 can modulate the anergy of T-cells and promote evasion of the immune system in UC [39]. Moreover, CTLA-4 can also assert its effects by removing CD80 molecules from neighboring APCs through trans endocytosis [33].

Although some studies have shown that blockade of CTLA-4 can lead to tumor regression in vitro using cell lines [31], the in vivo effects of CTLA-4 inhibition for UC have not been well studied. It is hypothesized that the inhibition of CTLA-4 can prevent anergy during the priming stage of T-cell activation, allowing for increased infiltration of lymphocytes into the tumor [40]. In addition, since CTLA-4 is also expressed in T-reg cells, blocking this immune checkpoint could reduce the ratio of T-reg to effector T-cells in the TME leading to better control of the tumor by the immune system [41]. There is currently no FDA-approved CTLA-4 antibody for bladder cancer, which highlights the need for additional research to assess CTLA-4 blockade in UC.

## 3. FDA-Approved Immune Checkpoint Inhibitors in Urothelial Carcinoma

Metastatic UC is initially responsive to chemotherapy, but has limited response duration and will often require second-line treatment upon recurrence [42]. The most common form of chemotherapy for UC is cisplatin-based chemotherapy; however, about two-thirds of patients are ineligible for this treatment [42]. Cisplatin is often cleared by the kidneys and is contraindicated in patients with kidney disease [42]. Historically, second-line treatment has had limited benefit for patients with progressive disease after chemotherapy [43]. Importantly, ICIs as a second-line therapy have shown the greatest benefit in terms of overall survival (OS) for UC [43]. The current FDA-approved drugs for metastatic UC are pembrolizumab, avelumab, and nivolumab.

### 3.1. Pembrolizumab

Pembrolizumab is a highly selective IgG4 humanized antibody that binds to PD-1 [44]. The KEYNOTE-045 phase III clinical trial demonstrated the efficacy of pembrolizumab in the treatment of metastatic UC [44]. It was an open-label, international trial including 542 patients with advanced urothelial cancer that recurred or progressed after chemotherapy [44]. The primary endpoints were OS and progression-free survival (PFS), which were assessed by calculating the tumor PD-1 ligand positive score, with 10% or more being the cutoff [44]. The cohort was divided into two groups in a 1:1 ratio with one group receiving 200 mg of pembrolizumab intravenously every three weeks. For comparison, the other group was given the investigator’s choice of paclitaxel, docetaxel, or vinflunine—all administered at 3 weeks [44]. To measure tumor size in response to therapy, the participants underwent tumor imaging at week nine and then every six weeks after that for a year and subsequently every 12 weeks. The median duration of study treatment in the pembrolizumab group was 3.5 months, while the chemotherapy group was treated for 1.5 months.

The results of the study showed a significantly higher OS rate for the pembrolizumab group with a hazard ratio of 0.73 [44]. The pembrolizumab group had a median OS of 10.3 months compared with the chemotherapy group with 7.4 months. Moreover, the estimated overall response rate in the pembrolizumab group was 43.9% compared with 30.7% in the chemotherapy group. There was no difference in the PFS between the pembrolizumab and the chemotherapy group. However, there was a higher objective response rate in the pembrolizumab group (21.1%) compared with the chemotherapy group (11.4%). In addition, the pembrolizumab group had fewer adverse effects. The most common treatment-related side effects were pruritus, fatigue, and nausea [44]. Based on these results, pembrolizumab received approval from the FDA for second-line neoadjuvant treatment of metastatic UC in the US. The tolerability and antitumor effect of pembrolizumab as first-line therapy was in the KEYNOTE-052 study, which showed a higher frequency of response to pembrolizumab for cisplatin-ineligible patients with locally advanced and unresectable or metastatic UC and high PD-L1 expression [45].

### 3.2. Nivolumab

Nivolumab is a humanized IgG4 antibody against PD-1 that was FDA-approved as a second-line treatment for metastatic UC based on the Checkmate 275 phase II clinical trial [46]. The Checkmate 275 was a multicenter, single-arm study of 270 patients, aged eighteen or older, with metastatic UC or unresectable local UC [46]. Patients received 3 mg of nivolumab every 2 weeks until disease progression, clinical deterioration, or unacceptable toxicity [46]. The primary endpoint was overall objective response and PD-L1 positive expression (≥5% and ≥1%) [46].

The overall objective response rate was 20% in the nivolumab group compared with 10% in the control group with no significant difference for PD-L1 expression between both groups. Importantly, there was increased response for patients with high PD-L1 expression. Treatment-related adverse effects occurred in 64% of the cohort with the most common adverse effect being fatigue, which was noted in 17% of the patients. In the cohort, 18% of patients experienced fatigue and diarrhea, 1% suffered from pneumonitis, 1% from pemphigoid, and <1% suffered from lung-related complications and autoimmune reactions like a pruritic rash [46].

Subsequently, nivolumab was approved for second-line treatment of metastatic or unresectable UC in the US. Since then, nivolumab has been approved as an adjuvant treatment for MIBC following radical surgical resection based on the results of CheckMate 274 [47]. This phase III, double-blind, randomized trial measured efficacy using disease-free survival (DFS) as its endpoint [47]. Median DFS in the nivolumab group was 20.8 months while the placebo group was 10.8 months. In addition, the median survival free from recurrence in the nivolumab group was 22.9 months compared with 13.7 months in the placebo group. In conclusion, nivolumab demonstrated longer DFS following radical surgery and was granted FDA approval in this setting.

### 3.3. Avelumab

Avelumab is an IgG1-type antibody against PD-L1 that prevents the binding of PD-L1 to PD-1 [48]. In 2020, avelumab received FDA approval for use as maintenance therapy in locally advanced or metastatic bladder cancer following the JAVELIN 100 trial [48]. This was an international, open-label phase III trial that lasted from May 2016 to June 2019 [48]. The inclusion criteria included locally advanced or metastatic bladder cancer patients who had stable disease after receiving four to six cycles of chemotherapy with gemcitabine and cisplatin or carboplatin [48]. Additionally, patients had to be treatment-free for at least four weeks before enrollment. Patients were randomly selected on a 1:1 ratio into the avelumab or control group. In the avelumab group, patients received 10 mg/kg of avelumab plus the best supportive care, while the control group received only supportive care [48]. For both groups, PD-L1 expression was assessed using the Ventana PD-L1 assay with a PD-L1-positive score assigned to tumor cells or immune cells staining for at least 25% PD-L [48]. Tumor samples that had 100% staining for PD-L1 on immune cells when less than 1% of the area had immune cells were also classified as positive. To measure the primary and secondary endpoints, the tumors were measured using RECIST. A total of 51.1% of patients had PD-L1-positive tumors with the breakdown being 57% in the avelumab group and 56.3% in the control group.

The primary endpoint was OS and the secondary endpoints were PFS and safety. The OS at 12 months was 71.3% in the avelumab group compared with 58.4% in the control group with the median OS being 21.4 months and 14.3 months, respectively. The PD-L1-positive group also demonstrated longer survival in the avelumab group at 79.1% compared with 60% in the control group. Moreover, PD-L1-negative patients in the avelumab group had an OS of 18.8 months compared with 13.7 months in the control group [48]. PFS and overall response rate were also markedly higher in the avelumab group compared with the control group. In the overall population, the avelumab group had a PFS of 3.7 months compared with 2.0 months in the control group. PD-L1-positive patients given avelumab had a median PFS rate of 5.7 months compared with 2.1 months in the control group, while PD-L1-negative patients in the avelumab group had a PFS rate of 3.0 months compared with 1.9 months in the control group. Therefore, PD-L1-positive patients performed significantly better than PD-L1-negative patients.

Adverse effects of any grade occurred in 98% of patients in the avelumab group compared with 77.7% in the control group, with the most common adverse effects being fatigue, pruritus, urinary tract infections, and diarrhea [48]. Adverse events of grade 3 or higher occurred in 47.4% of the avelumab group with the control group having 25.2%. The most common adverse effect in the avelumab group was anemia, urinary tract infections, fatigue, and hematuria. Additionally, 29.4% of patients in the avelumab group had an immune-related adverse effect with the most common being thyroid disorders [48]. Due to the higher OS and PFS rates in the avelumab group, it was granted FDA approval for maintenance therapy in locally advanced or metastatic bladder cancer. The key clinical trials leading to FDA approval of ICIs are presented in Table 1.

## 4. ICI Combination Therapies

The use of ICIs in UC is instrumental in the activation of tumor-infiltrating lymphocytes against the tumor. Moreover, tumor-infiltrating lymphocytes interface with other immune cells and cytokines in the tumor microenvironment [9]. Due to the crosstalk between these different elements, there will be indirect effects of ICIs on the immune microenvironment. ICIs reactivate T-cells and increase IFNγ in the environment. Although IFNγ upregulates PD-L1 expression, it is involved in many of the inflammatory processes of the body [22]. IFNγ is involved in the differentiation of monocytes into the M1 macrophage lineage, also known as the pro-inflammatory subtype of macrophages [22]. In this way, ICIs not only influence lymphocytes but can indirectly prime other immune cells to decrease tumor volume. The upregulation of PD-L1 due to IFNγ is addressed by PD-L1 inhibitors; therefore, the anti-tumor effects of IFNγ can be achieved while avoiding the pro-tumor effects. Unfortunately, only a few patients have durable long-term responses to ICI therapy [49]. The potential for ICIs in UC can be transformational if the mechanisms behind resistance are understood and addressed. A possible way to address this resistance is combination therapy. The use of PD-1/PD-L1 inhibitors with CTLA-4 inhibitors could address resistance by targeting different pathways simultaneously.

Studies have shown that reactivation of T-cells under PD-1 inhibition potentially stimulates compensatory immune checkpoints, such as CTLA-4 [30]. CTLA-4 can induce anergy of T-cells in the priming stage by binding CD80 on APCs. Targeting CTLA-4 at the priming stage would allow for increased de novo T-cell synthesis and proliferation [30]. In addition to reactivating existing T-cells using PD-1/PD-L1 inhibitors, CTLA-4 adds the benefit of increasing the number of T-cells in the TME. Therefore, disruption of multiple pathways for immune checkpoints in UC can lead to prolonged activation of T-cells. This can be achieved with combination therapy. However, the use of CTLA-4 for UC has been limited even though it is approved for use in the treatment of other highly mutagenic cancers, such as melanoma, renal cell carcinoma (RCC), and non-small cell lung cancer (NSCLC). In 2010, the CTLA-4 antibody ipilimumab was approved for the treatment of melanoma following a successful phase III trial [50]. Combination therapy of ipilimumab and PD-1 inhibitors was subsequently approved for the treatment of melanoma [51], NSCLC [52], and RCC [53].

### 4.1. Approved Combination Therapies in Other Cancers

The checkmate 9LA clinical trial tested the use of nivolumab and ipilimumab combined with chemotherapy (experimental) vs. chemotherapy alone (control) for stage IV or recurrent NSCLC. The dosage for nivolumab was 360 mg intravenously every 3 weeks, and ipilimumab was administered at a dose of 1 mg/kg. This study used OS as its primary endpoint. Importantly, the experimental group had a longer survival at 14.1 months compared with 10.7 months in the control group [52]. There were high-grade adverse events in both groups, with seven recorded deaths in the experimental group and six in the control group. Nevertheless, this combination was approved after showing positive OS results.

Checkmate 214 was a phase III clinical trial of nivolumab plus ipilimumab for untreated advanced RCC. This trial tested nivolumab plus ipilimumab combination (experimental) vs. sunitinib, a multi-tyrosine kinase inhibitor (control) [53]. The experimental group received 3 mg/kg of nivolumab plus 1 mg/kg of ipilimumab for three weeks for four doses followed by 3 mg/kg of nivolumab every 2 weeks. At 18 months, the experimental group had significantly higher OS at 75% compared with 60% in the control group. PFS was also higher in the experimental group at 11.6 months. Grade 3–4 adverse effects occurred in 46% with a 22% drop rate due to treatment-related side effects. However, the safety profile and survival benefit were favorable enough for FDA approval.

Checkmate 067 was a phase III clinical trial that tested nivolumab plus ipilimumab (experimental) vs. nivolumab or ipilimumab alone in stage III or IV melanoma [51]. The experimental group received 1 mg/kg of nivolumab plus 3 mg/kg of ipilimumab every three weeks for four doses followed by 3 mg/kg of nivolumab every 2 weeks. Following the four-year follow-up, nivolumab plus ipilimumab provided a durable and sustained survival benefit with a good safety profile for patients with melanoma [51]. In the experimental group, 59% of patients experienced treatment-related grade 3–4 adverse effects, with the most common grade 3 side effect being diarrhea and the most common grade 4 effect being increased lipase. In 2019, a phase IIIb/IV clinical trial evaluated a lower dose of ipilimumab in advanced melanoma [54]. Patients who were given 3 mg/kg/d of nivolumab plus 1 mg/kg/d of ipilimumab had a lower incidence of grade 3–4 adverse effects compared with patients given 1 mg/kg/d of nivolumab plus 3 mg/kg/d of ipilimumab, with no significant difference in efficacy between the two doses. Table 2 summarizes the findings of approved combination therapies.

### 4.2. Clinical Studies in UC

The previously mentioned studies demonstrated similar safety profiles with the use of the same ipilimumab dose (1 mg/kg). Although treatment-related side effects were present in the experimental groups, the risk-to-benefit ratio was still favorable. However, the same has not been observed in UC. It is unclear why it has been difficult to use CTLA-4 in clinical treatment for UC even though it is highly expressed. One hypothesis is that an increased dosage of CTLA-4 is needed to see benefits leading to severe treatment-related adverse effects. From 2015 to 2017, AstraZeneca conducted the DANUBE trial. This phase three trial compared the PD-L1 antibody durvalumab combined with the CTLA-4 antibody tremelimumab vs. chemotherapy alone in UC [55]. The clinical trial was unsuccessful in reaching its primary endpoint of OS. A phase I/II trial testing nivolumab and ipilimumab emerged, and while it did not reach phase III, the results looked promising [56]. The trial divided patients on a 1:1:1 ratio into three groups: 3 mg of nivolumab, 3 mg of nivolumab plus 1 mg of ipilimumab, and 1 mg of nivolumab plus 3 mg of ipilimumab [56]. The most significant increase in ORR was seen in the group given 3 mg of ipilimumab with the safety profile being unfavorable. This highlights the potential of CTLA-4 inhibition in patients with UC who can tolerate treatment.

Another potential roadblock to the combined inhibition of multiple immune checkpoints could be the upregulation of other checkpoint receptors within UC tumor cells. For example, lymphocyte activation gene 3 (LAG3) is another immune checkpoint protein expressed on T-cells that leads to the exhaustion of activated T-cells [57]. LAG3 binds to MHC II as well as other receptors, such as fibrinogen-like protein 1 (FGL-1) [57,58]. When LAG3 binds to its receptors, it decreases the release of cytokines and granzymes and negatively regulates CD8+ T-cell proliferation [57]. It also promotes differentiation into T-reg cells, thereby promoting an anti-inflammatory environment within tumors [59]. A study conducted by Zeng et al. analyzed the survival rates of patients with MIBC who expressed intraepithelial vs. stromal LAG3 [60]. They found that LAG3 was preferentially expressed in the stroma and associated with decreased 5-year survival [60]. Patients with MIBC who expressed high levels of stromal LAG3 also had increased infiltration of T-reg cells within the TME with a concurrent increase in anti-inflammatory cytokines such as IL-10 and TGFβ [60]. The utilization of LAG3 in metastatic UC has not been elucidated fully in the literature. Another study conducted by Vanguri et al. investigated the use of LAG3 as a biomarker of response to anti-PD-1/PD-L1 in metastatic and primary UC [61]. Their study found that LAG3 was a better biomarker of response to anti-PD-1/PD-L1 therapy, especially when combined with PD-L1 expression. Patients who responded had higher expression of LAG3 compared with non-responders [61]. These findings could point to a response of the TME within UC to blockade PD-1 and PD-L1. Exploration of the expression of LAG3 in metastatic UC would be important to illuminate potential mechanisms of resistance to anti-PD-1/PD-L1 and anti-CTLA-4 combinations in UC.

In addition to LAG3, there are other immune checkpoints that have been discovered to play a role in UC, such as the T-cell immunoglobulin domain and mucin domain-containing molecule (TIM-3) and the T-cell immunoreceptor tyrosine-based inhibitory motif domain (TIGIT). TIM3 and TIGIT are expressed on CD4^+^ and CD8^+^ T-cells as well as T-reg and innate immune cells [62]. Analysis of peripheral blood mononuclear cells from patients with UC showed increased expression of TIM3 and TIGIT on T-cells and natural killer cells [62]. Thus, further research is needed to understand the cross-talk between multiple immune checkpoint pathways to improve ICI combination response while minimizing toxicity.

### 4.3. Future Directions and Controversies

Further studies are needed to examine relevant biomarkers and other immune checkpoints. When PD-1 inhibitors were being developed, it was believed that PD-L1 expression levels would be a biomarker for predicting patients that would benefit from PD-1/PD-L1 inhibition therapy [30]. There are mixed results on the expression of PD-L1 as a marker for clinical response. In addition, manufacturers use non-standardized assays to measure PD-L1 expression with differing thresholds of positivity. For example, Ventana classifies ≥ 5% of PD-L1 expression as positive and was used for atezolizumab [30]. However, pembrolizumab used the IHC 22C3 PharmDx assay, which classifies ≥ 10% as PD-L1 positive [44]. Similarly, nivolumab and avelumab used different assays. Therefore, no standardized threshold for PD-L1-positive tumors has been established making any comparative correlations difficult. The only consistent finding has been that both PD-L1-negative and -positive tumors respond to PD-1/PD-L1 for reasons that are still unknown [63,64]. PD-L1 expression is also limited as a biomarker in combination therapy as other ICIs do not specifically target PD-L1. The density of lymphocytes in the TME could be a possible biomarker since PD-1, PD-L1, and CTLA-4 inhibitors have all been shown to increase CD4 and CD8 cell infiltration [65].

Genetic signatures such as molecular subtypes and TMB are other prospective biomarkers. The molecular subtype classification of UC carried out by TCGA has been used in some clinical trials as a possible biomarker [18]. Nivolumab reported response rates based on molecular subtypes in the CheckMate 275 trial. Basal 1 had the highest response rate followed by luminal cluster 2 [46]. Recent data also suggest that TMB might be a more robust biomarker than PD-L1 expression in UC [66]. It is hypothesized that higher titers of neoantigens present in patients indicate a more robust response by the immune system. Additionally, those with higher TMB had a better response rate as opposed to those with lower TMB [46]. However, TMB has not been formally vetted as a biomarker in phase III studies, and there is no defined threshold to indicate high vs. low TMB levels [66].

Other molecular players involved in tumor promotion need to be investigated for future drug development. CTLA-4 is not the only compensatory checkpoint upregulated during PD-1/PD-L1 resistance. Other checkpoints include TIM-3, TIGIT, and LAG3. TIM-3 is expressed on CD4^+^ T-cells and CD8^+^ T-cells and it binds to different receptors, including galectin-9, CEACAM-1, and HMGB-1 [30]. The binding of TIM-3 to its receptors leads to defective expression of proinflammatory cytokines and it signals the use of a different mechanism of action from PD-1/PD-L1. LAG3 is expressed on multiple cells, most notably T-cells and T-regs [57,67]. LAG3 is highly expressed in T-cells after activation, but the role of LAG3 in immune regulation is unclear. The main receptor for LAG3 is MHC-II [67]. However, it may also bind to other receptors including galectin-9 and FGL-1 [57,67].

There are also other potential targets that can be used in combination with ICIs as treatment such as Nectin-4. Nectin-4 is a tumor antigen expressed on most UC cells and the basis for the development of the drug enfortumab vedotin [68]. Enfortumab vedotin is an antibody-drug conjugate that consists of an antibody against Nectin-4 and a microtubule-disrupting agent [69]. In 2023, a recent phase III trial of enfortumab vedotin in combination with pembrolizumab vs. chemotherapy for untreated locally advanced metastatic UC was conducted [69]. The use of enfortumab vedotin in combination with pembrolizumab nearly doubled the median PFS and OS of patients compared with OS [69].

FGFR3 is another novel target that has been explored within UC. FGFR3 is a tyrosine kinase that is mutated in 20% of UC leading to oncogenesis [70]. In 2019, erdafitinib was the first FGFR3 inhibitor approved for locally advanced or metastatic UC with FGFR2/3 mutations who progressed on prior chemotherapy [71]. Investigations into a combination of FGFR2/3 inhibitors with other treatment modalities could abrogate possible resistance that can arise in monotherapy. Further evaluations into the synergism of ICIs and FGFR2/3 inhibitors are warranted in the treatment landscape of UC.

## 5. Conclusions

The targeting of immune checkpoints in UC has been demonstrated to be a promising therapeutic approach, but further studies are needed to develop biomarkers and to understand mechanisms of resistance. The crosstalk between immune cells in the TME has shown how PD-L1 is upregulated on both tumor cells and dendritic cells. Simultaneously, comprehensive flow cytometry studies have determined that CTLA-4 upregulation may contribute to PD-1/PD-L1 antibody resistance. This observation is poorly understood, indicating that additional preclinical studies are needed. The data suggest that upregulation of CTLA-4 during PD-1/PD-L1 treatment may promote drug resistance, indicating that combination therapy should be evaluated. These approaches may help increase survival in patients with UC undergoing ICI therapy.

## Figures and Tables

**Figure 1 cancers-16-00131-f001:**
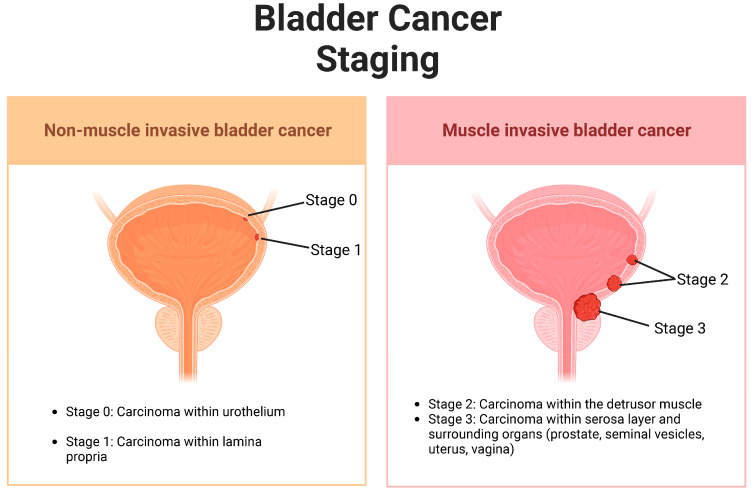
NMIBC vs. MIBC. Not included in this figure is stage 4 bladder cancer, in which the cancer metastasizes to the lymph nodes and distal organs [5]. Illustration created with BioRender.com (accessed on 30 November 2023).

**Figure 2 cancers-16-00131-f002:**
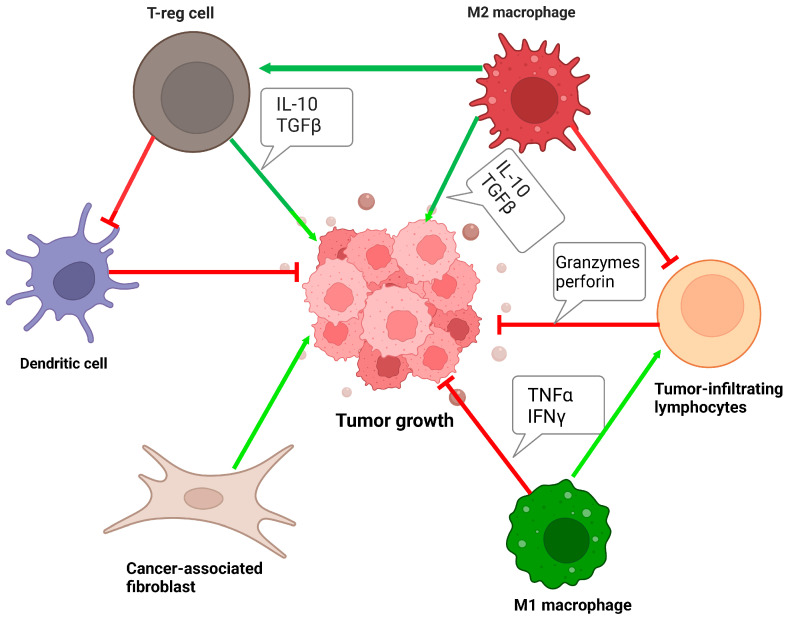
Interactions between different cells in the tumor microenvironment of UC. The green arrow represents promotion and the red arrow represents inhibition. Illustration created with BioRender.com (accessed on 30 November 2023).

**Figure 3 cancers-16-00131-f003:**
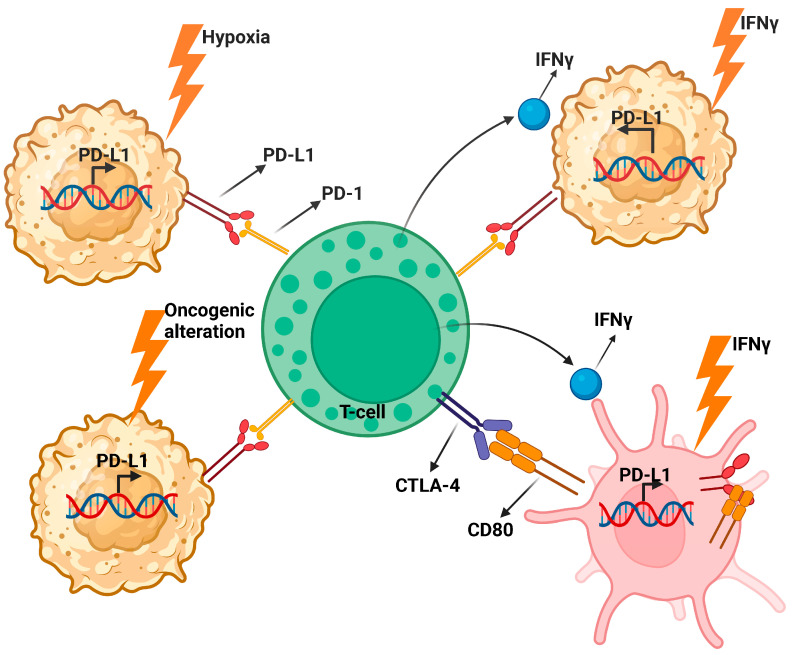
Mediators of PD-L1 transcription in UC. Included are the different pathways by which PD-L1 expression is upregulated both in tumor cells and antigen-presenting cells. Illustration created with BioRender.com (accessed on 30 November 2023).

**Table 1 cancers-16-00131-t001:** Overview of clinical trials for current FDA-approved ICIs for metastatic UC.

Treatment	Name and Year	Phase	Line of Therapy	N	Findings
Pembrolizumab vs. chemotherapy	Keynote-045, 2017	III	Second	542	OS: 10.3 vs. 7.4 months (all patients) OS: 8 vs. 5.2 months (for PD-L1 ≥ 10%)
Nivolumab post platinum-based chemotherapy	CheckMate 275, 2017	II	Second	270	ORR: 19.6% (52/265)
Avelumab maintenance vs. best supportive care	Javelin Bladder 100, 2020	III	First	700	OS: 21.4 vs. 14.3 months

ORR: overall response rate; OS: overall survival.

**Table 2 cancers-16-00131-t002:** Summary of clinical trials of PD-1/PD-L1 and CTLA-4 inhibition combination therapy applied on RCC, melanoma, NSLC, and UC.

Cancer Type	Treatment	Phase	Findings	Adverse Effects
NSLC—stage IV/recurrent	Nivolumab (360 mg IV) + ipilimumab (1 mg/kg) + chemotherapy vs. chemotherapy alone	III	OS: 14.1 vs. 10.7 months	Grade 3–4: 47% vs. 38%
RCC—advanced stage	Nivolumab (3 mg/kg) + ipilimumab (1 mg/kg) vs. sunitinib	III	ORR: 42% vs. 27%	Grade 3–4: 46% vs. 63%
Melanoma—stages III and IV	Nivolumab (1 mg/kg) + ipilimumab (3 mg/kg) vs. nivolumab alone (3 mg/kg) vs. ipilimumab alone (3 mg/kg)	III	PFS: 11.5 vs. 6.9 vs. 2.9 months	Grade 3–4: 59% vs. 22% vs. 28%
Melanoma—advanced stage	Nivolumab (3 mg/kg) + ipilimumab (1 mg/kg) vs. nivolumab (1 mg/kg) + ipilimumab (3 mg/kg)	IIIb/IV	ORR: 50.6% vs. 48%PFS: 9.9 months vs. 8.9 months	Grade 3–5: 34% vs. 48%
Urothelial carcinoma—metastatic	Nivolumab (3 mg/kg) vs. nivolumab (3 mg/kg) + ipilimumab (1 mg/kg) vs. nivolumab (1 mg/kg) + ipilimumab (3 mg/kg)	I/II	ORR: 25.6% vs. 26.9% vs. 38%	Grade 3–4: 26.9% vs. 30.8% vs. 39.1%

OS: overall survival; ORR: overall response rate; PFS: progression-free survival; RCC: renal cell carcinoma; and NSLC: non-small cell lung cancer.

## Data Availability

No new data were created or analyzed in this study. Data sharing is not applicable to this article.

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
