# Peer review of "Current Landscape of Immune Checkpoint Inhibitors for Metastatic Urothelial Carcinoma: Is There a Role for Additional T-Cell Blockade?†"

_cancers, 2023, doi:10.3390/cancers16010131_

Round 1

Reviewer 1 Report

Comments and Suggestions for Authors

This is a very well written review and is  significant contribution summarizing ICI in bladder cancer. 

I have only one minor correction :spelling mistakeline 174 ... replace damper to "dampen"

Thanks much

Author Response

Response: Thank you for the positive feedback regarding our manuscript. We have made the suggested correction to the spelling on “dampen”.

Reviewer 2 Report

Comments and Suggestions for Authors

The review article entitled ‘Current Landscape of Immune Checkpoint Inhibitors for Metastatic Urothelial Carcinoma: Is There a Role for Additional T-cell Blockade?’ Was well received.

Please use uniform abbreviations/names throughout manuscript. For example, cytotoxic T lymphocytes are abbreviated as both CD8 cells or CD8 T cells in manuscript. It should be CD8+ T cells, as this is most common name and will not cause confusion in the mind of reader.

By what mechanism the proposed combined anti-PD1 and anti-CTLA4 would be beneficial. Some papers have reported that blocking one or two types of checkpoint receptors would lead to upregulation of other checkpoints like LAG-3 or KLRG1. The authors must give some perspective on this issue

Why is the status of UC for expressing ligands for other checkpoint receptors?

The quality of figures is not good and needs significant improvement 

Comments on the Quality of English Language

fine

Author Response

The review article entitled ‘Current Landscape of Immune Checkpoint Inhibitors for Metastatic Urothelial Carcinoma: Is There a Role for Additional T-cell Blockade?’ Was well received.

Please use uniform abbreviations/names throughout manuscript. For example, cytotoxic T lymphocytes are abbreviated as both CD8 cells or CD8 T cells in manuscript. It should be CD8+ T cells, as this is most common name and will not cause confusion in the mind of reader.

Response: Thank you for this suggestion. We modified the text to use uniform abbreviations in the manuscript and corrected the abbreviation to CD4+ T-cells and CD8+ T-cells, respectively.

By what mechanism the proposed combined anti-PD1 and anti-CTLA4 would be beneficial. Some papers have reported that blocking one or two types of checkpoint receptors would lead to upregulation of other checkpoints like LAG-3 or KLRG1. The authors must give some perspective on this issue

Response: We have further expanded our discussion of other checkpoint inhibitors that can be upregulated in urothelial carcinoma including LAG3, TIM3 and TIGIT. Our manuscript proposed the benefit of CTLA-4 inhibition at the priming stage of T-cell activation combined with PD-1/PD-L1 inhibition for already activated T-cells.

Why is the status of UC for expressing ligands for other checkpoint receptors?

Response: We have expanded our discussion to include more information on various checkpoint receptors including LAG3, TIM3 and TIGIT.

The quality of figures is not good and needs significant improvement 

Response: Thanks for this comment. We used moderate quality jpegs for the figures during the initial review stage. We have now included TIFFs with higher resolution to improve the quality. We will work with the Cancers Editorial Staff to provide additional figure files if needed.

Reviewer 3 Report

Comments and Suggestions for Authors

The review article titled “Current Landscape of Immune Checkpoint Inhibitors for Metastatic Urothelial Carcinoma: Is There a Role for Additional T cell Blockade” by Vanessa Ogbuji et al. explores the landscape of the current immune checkpoint inhibition (ICI) treatment in urothelial carcinoma (UC). This article is a reproduction of the master dissertation at the University of Arizona by the first author (Chizitaram Vanessa Ogbuji) titled "Combination of Immune Checkpoint Inhibitors in Metastatic Urothelial Carcinoma: PD-1/PD-L1 with CTLA-4".

The reviewer appreciates the efforts by the authors, given the wide coverage of ICI treatment in multiple aspects, which certainly makes it a nicely written master dissertation.

Before giving my comments, the reviewer would like to point to an editorial written by Warren Chan in ACS Nano titled “Writing Excellent Review Articles” (https://doi.org/10.1021/acsnano.3c00497). It outlines the features of an excellent review and traits of a poor one. As the editorial says, the authors should not “rewriting a line from the paper’s abstract or inaccurately describe the original study’s findings, or both. These problems can derail a field” but focus to “guides the field of research”.

Given the quality of this article, the reviewer does not recommend acceptance for publication in current form. This review is largely a copy of information that has been presented in other places (other articles, reviews and books) and reorganized. This review fails to present a new insightful perspective, contrary to the authors’ claim.

1.       Even though the authors acknowledge that “UC can be divided into 3 categories: non muscle invasive bladder cancer (NMIBC), muscle invasive bladder cancer (MIBC), and metastatic UC” and the title indicate the “Metastatic Urothelial Carcinoma” as the focus of this review. However, throughout this review, most of the description and discussion are on UC in general, not metastatic UC.

2.       Even if the authors want to focus on ICI treatment on UC, the readers’ excitement would quickly diminish as this topic is being covered extensively by numerous review articles, books, and conferences. The reviewer does not find anything particularly new which has not been uncovered by existing literature.  

3.       Even though the authors attempt to address the question “Is There a Role for Additional T cell Blockade”, the answers to this question (mostly in section 4.3) are very superficial. Should there be drugs that target multiple immune checkpoints, or are there other checkpoints that should be targeted? The answer to why “there are limited investigations evaluating the combination of CTLA 4 and PD 1/PD L1 blockade” should be very straightforward if the authors have done more diligent literature research, however, even such a simple “why” question is inadequately addressed. 

4.  Throughout this review, most of the contents are applicable to a wide range of cancers, not just UC, including but not limited to most of the sections except for section 4.2 (which counts for less than 10% of the entire article). The uniqueness of metastatic UC is not sufficiently presented.

Overall, the reviewer congratulates the authors for summarizing multiple aspects of UC, including TME, preclinical research, clinical trials into one cohesive article. However, the reviewer, who is an active researcher on ICI mechanism of action, was disappointed to find out there is very little new knowledge being presented. The lack of significant contribution to the scientific and clinical fields undermines the value of this review. 

Author Response

Reviewer #3 (Reviewer Comments to the Author):

The review article titled “Current Landscape of Immune Checkpoint Inhibitors for Metastatic Urothelial Carcinoma: Is There a Role for Additional T cell Blockade” by Vanessa Ogbuji et al. explores the landscape of the current immune checkpoint inhibition (ICI) treatment in urothelial carcinoma (UC). This article is a reproduction of the master dissertation at the University of Arizona by the first author (Chizitaram Vanessa Ogbuji) titled "Combination of Immune Checkpoint Inhibitors in Metastatic Urothelial Carcinoma: PD-1/PD-L1 with CTLA-4".

The reviewer appreciates the efforts by the authors, given the wide coverage of ICI treatment in multiple aspects, which certainly makes it a nicely written master dissertation.

Before giving my comments, the reviewer would like to point to an editorial written by Warren Chan in ACS Nano titled “Writing Excellent Review Articles” (https://doi.org/10.1021/acsnano.3c00497). It outlines the features of an excellent review and traits of a poor one. As the editorial says, the authors should not “rewriting a line from the paper’s abstract or inaccurately describe the original study’s findings, or both. These problems can derail a field” but focus to “guides the field of research”.

Given the quality of this article, the reviewer does not recommend acceptance for publication in current form. This review is largely a copy of information that has been presented in other places (other articles, reviews and books) and reorganized. This review fails to present a new insightful perspective, contrary to the authors’ claim.

Response: We appreciate this reviewer’s perspective regarding his/her preferred format and content in a well written review article. However, it is our view that a strong review article should serve as a resource for the non-expert to quickly become familiar with the current landscape of a particular field. Those who are already renowned subject matter experts do not need to read a review to appreciate the current status of their respective field.  Indeed, we feel that it is not an expectation for a review article to break new ground and our article was not written with that purpose in mind. The focus of our article is to review the current state of the field of the use of immune checkpoint inhibitors in urothelial carcinoma. Our review summarized key studies and mentions important findings related to the topic. We feel that this goal was accomplished and that following the completion of the suggested revisions recommended by the reviewers, that this article will be of high interest to the readership of Cancers.

  1. Even though the authors acknowledge that “UC can be divided into 3 categories: non muscle invasive bladder cancer (NMIBC), muscle invasive bladder cancer (MIBC), and metastatic UC” and the title indicate the “Metastatic Urothelial Carcinoma” as the focus of this review. However, throughout this review, most of the description and discussion are on UC in general, not metastatic UC.

Response: Thank you for this comment. Our objective was to provide a review of immune checkpoint inhibitors in the primary UC and metastatic setting. The clinical studies discussed in this review were focused on current FDA approved immune checkpoint inhibitors for metastatic UC given that the target audience is for clinicians and trainees. Since immunotherapy is a systemic treatment, most of its use is for both locally advanced and metastatic disease. Based on your helpful feedback, we included new information on current clinical trials of other molecular targets within urothelial carcinoma such as Nectin-4 and FGFR3 in the revised manuscript.

  1. Even if the authors want to focus on ICI treatment on UC, the readers’ excitement would quickly diminish as this topic is being covered extensively by numerous review articles, books, and conferences. The reviewer does not find anything particularly new which has not been uncovered by existing literature.  

Response: We agree with the reviewer that the topic of ICI in UC and other malignancies is a “hot” area, which has resulted in the publication of some other reviews covering this general area. Despite that, we feel that our article adds key information summarizing the rationale for using ICI in UC management while also describing the most recent advances in the field. Given that, we believe that our manuscript would be of significant interest to a broad audience by bringing this important information together into one concise manuscript.

  1. Even though the authors attempt to address the question “Is There a Role for Additional T cell Blockade”, the answers to this question (mostly in section 4.3) are very superficial. Should there be drugs that target multiple immune checkpoints, or are there other checkpoints that should be targeted? The answer to why “there are limited investigations evaluating the combination of CTLA 4 and PD 1/PD L1 blockade” should be very straightforward if the authors have done more diligent literature research, however, even such a simple “why” question is inadequately addressed. 

Response: Thanks for sharing your perspective, which gave us an opportunity to strengthen this specific aspect of our review. Based on your helpful suggestion, we have added significant new information to address this point and have also included more discussion on other immune checkpoint targets to improve ICI therapy. We have also highlighted the current working hypothesis in the field that combination therapy strategies using existing FDA approved immune checkpoint inhibitors provides a multifactorial approach to treatment with the potential to significantly improve patient outcomes. We also mentioned the associated toxicities that have been reported and the limited preclinical research into the underlying cause. Another section was incorporated into the revised manuscript discussing possible upregulation of other immune checkpoints in response to combination therapy.

  1.  Throughout this review, most of the contents are applicable to a wide range of cancers, not just UC, including but not limited to most of the sections except for section 4.2 (which counts for less than 10% of the entire article). The uniqueness of metastatic UC is not sufficiently presented.

Response: Our objective was to not only discuss the state of ICI therapy in UC, but also to describe the use of ICI in other tumor types and how these approaches may also be applicable to UC. We have included additional information to stress this point in the revised manuscript.

Overall, the reviewer congratulates the authors for summarizing multiple aspects of UC, including TME, preclinical research, clinical trials into one cohesive article. However, the reviewer, who is an active researcher on ICI mechanism of action, was disappointed to find out there is very little new knowledge being presented. The lack of significant contribution to the scientific and clinical fields undermines the value of this review. 

Response: We acknowledge that the reviewer is an expert in ICI therapy and thus, some of the information may be well known to him/her. Our objective was to provide a comprehensive literature review of the use of ICI in UC and the potential for future combination therapy for an audience with less subject matter expertise than this reviewer. We feel that we have achieved this goal and that the review will be informative for many investigators that are new to the field.

Round 2

Reviewer 2 Report

Comments and Suggestions for Authors

The paper is improved sufficiently. 

Reviewer 3 Report

Comments and Suggestions for Authors

Given the minor changes since the last version and the authors’ refusal of significant changes, the reviewer here does not recommend acceptance for publication given the current form. The major issues I previously pointed out were not adequately addressed.

With all respects, the reviewer here does not agree with the following argument in the authors’ response: “However, it is our view that a strong review article should serve as a resource for the non-expert to quickly become familiar with the current landscape of a particular field. Those who are already renowned subject matter experts do not need to read a review to appreciate the current status of their respective field.  Indeed, we feel that it is not an expectation for a review article to break new ground and our article was not written with that purpose in mind.” I would encourage whoever wrote this to talk to the editor-in-chief of any renowned journals (not special issue editor from MDPI) and see his/her reaction. I shared an example from Warren Chan in ACS Nano already.

Back to my specific comments previously,

(1) The authors did not address the different ICI strategies between metastatic UC and vs NMIBC and/or MIBC, despite the title of this article.

(2) The authors did not fully address the uniqueness of UC when it comes to the administration of ICI, both mechanistically and clinically, from other type of cancer

(3)  The descriptions/arguments on combinatory approach and new targets are still very superficial.

I will endorse this article, if the authors can point out at least 5 new findings or forward-looking conclusions that have not been covered in the following review articles in recent years in urothelial carcinoma & bladder cancer.

·         Bidnur, S., Savdie, R., & Black, P. C. (2016). Inhibiting immune checkpoints for the treatment of bladder cancer. Bladder Cancer, 2(1), 15-25.

·         Hanna, K. S. (2017). A review of immune checkpoint inhibitors for the management of locally advanced or metastatic urothelial carcinoma. Pharmacotherapy: The Journal of Human Pharmacology and Drug Therapy, 37(11), 1391-1405.

·         Hsu, F. S., Su, C. H., & Huang, K. H. (2017). A comprehensive review of US FDA-approved immune checkpoint inhibitors in urothelial carcinoma. Journal of Immunology Research, 2017.

·         Siefker-Radtke, A., & Curti, B. (2018). Immunotherapy in metastatic urothelial carcinoma: focus on immune checkpoint inhibition. Nature Reviews Urology, 15(2), 112-124.

·         Cheng, W., Fu, D., Xu, F., & Zhang, Z. (2018). Unwrapping the genomic characteristics of urothelial bladder cancer and successes with immune checkpoint blockade therapy. Oncogenesis, 7(1), 2.

·         Tzeng, A., Diaz-Montero, C. M., Rayman, P. A., Kim, J. S., Pavicic, P. G., Finke, J. H., ... & Grivas, P. (2018). Immunological correlates of response to immune checkpoint inhibitors in metastatic urothelial carcinoma. Targeted Oncology, 13, 599-609.

·         Siefker-Radtke, A. O., Apolo, A. B., Bivalacqua, T. J., Spiess, P. E., & Black, P. C. (2018). Immunotherapy with checkpoint blockade in the treatment of urothelial carcinoma. The Journal of urology, 199(5), 1129-1142.

·         Kim, H. S., & Seo, H. K. (2018). Immune checkpoint inhibitors for urothelial carcinoma. Investigative and Clinical Urology, 59(5), 285-296.

·         Powles, T., & Morrison, L. (2018). Biomarker challenges for immune checkpoint inhibitors in urothelial carcinoma. Nature Reviews Urology, 15(10), 585-587.

·         Gopalakrishnan, D., Koshkin, V. S., Ornstein, M. C., Papatsoris, A., & Grivas, P. (2018). Immune checkpoint inhibitors in urothelial cancer: recent updates and future outlook. Therapeutics and clinical risk management, 1019-1040.

·         Massari, F., Di Nunno, V., Cubelli, M., Santoni, M., Fiorentino, M., Montironi, R., ... & Ardizzoni, A. (2018). Immune checkpoint inhibitors for metastatic bladder cancer. Cancer treatment reviews, 64, 11-20.

·         Alhalabi, O., Shah, A. Y., Lemke, E. A., & Gao, J. (2019). Current and Future Landscape of Immune Checkpoint Inhibitors in Urothelial Cancer. Oncology (08909091), 33(1).

·         Roviello, G., Catalano, M., Nobili, S., Santi, R., Mini, E., & Nesi, G. (2020). Focus on biochemical and clinical predictors of response to immune checkpoint inhibitors in metastatic urothelial carcinoma: Where do we stand?. International Journal of Molecular Sciences, 21(21), 7935.

·         Parikh, M., & Powles, T. (2021). Immune checkpoint inhibition in advanced bladder and kidney cancer: responses and further management. American Society of Clinical Oncology Educational Book, 41, e182-e189.

·         Chen, X., Chen, H., He, D., Cheng, Y., Zhu, Y., Xiao, M., ... & Cao, K. (2021). Analysis of tumor microenvironment characteristics in bladder cancer: implications for immune checkpoint inhibitor therapy. Frontiers in immunology, 12, 672158.

·         Roviello, G., Catalano, M., Santi, R., Palmieri, V. E., Vannini, G., Galli, I. C., ... & Nesi, G. (2021). Immune checkpoint inhibitors in urothelial bladder cancer: State of the art and future perspectives. Cancers, 13(17), 4411.

·         Lopez-Beltran, A., Cimadamore, A., Blanca, A., Massari, F., Vau, N., Scarpelli, M., ... & Montironi, R. (2021). Immune checkpoint inhibitors for the treatment of bladder cancer. Cancers, 13(1), 131.

·         Maiorano, B. A., De Giorgi, U., Ciardiello, D., Schinzari, G., Cisternino, A., Tortora, G., & Maiello, E. (2022). Immune-checkpoint inhibitors in advanced bladder cancer: Seize the day. Biomedicines, 10(2), 411.

·         Houssiau, H., & Seront, E. (2022). The evolution of immune checkpoint inhibitors in advanced urothelial carcinoma. Cancers, 14(7), 1640.

·         Zhu, A., Garcia, J. A., Faltas, B., Grivas, P., Barata, P., & Shoag, J. E. (2023). Immune Checkpoint Inhibitors and Long-term Survival of Patients With Metastatic Urothelial Cancer. JAMA Network Open, 6(4), e237444-e237444.

·         Okobi, T. J., Uhomoibhi, T. O., Akahara, D. E., Odoma, V. A., Sanusi, I. A., Okobi, O. E., ... & Harry, N. M. (2023). Immune Checkpoint Inhibitors as a Treatment Option for Bladder Cancer: Current Evidence. Cureus, 15(6).

Finally, I would encourage the authors that in order to impress the readers, new discoveries, novel approaches or insightful perspectives are more significant, other than reiterating what have been covered extensively.